# Correction for the Partial Volume Effects (PVE) in Nuclear Medicine Imaging: A Post-Reconstruction Analytic Method

Fabio Di Martino [1], Patrizio Barca [2], Eleonora Bortoli [1,*], Alessia Giuliano [3] and Duccio Volterrani [4]

1  Medical Physics Unit, University Hospital of Santa Chiara, 56126 Pisa, Italy; f.dimartino@ao-pisa.toscana.it
2  Department of Medical Physics, IRCCS AOUB, 40138 Bologna, Italy; barca.patrizio@gmail.com
3  Medical Physics Unit, Hospital of San Luca, 55100 Lucca, Italy; alessia.giuliano@uslnordovest.toscana.it
4  Nuclear Medicine Unit, University Hospital of Santa Chiara, 56126 Pisa, Italy; duccio.volterrani@med.unipi.it
*  Correspondence: eleonora.bortoli@gmail.com; Tel.: +39-3408764238

**Featured Application: The proposed approach allows recovering the loss of quantification due to Partial Volume Effect. This is an alternative post-processing method to the Recovery Coefficient method, with the same limitations, but with the advantage of deriving theoretically the dependencies of the parameters that describe the effect. With the proposed approach, the need of measuring these parameters every time they change could be overcome.**

**Abstract:** Quantitative analyses in nuclear medicine are increasingly used, both for diagnostic and therapeutic purposes. The Partial Volume Effect (PVE) is the most important factor of loss of quantification in Nuclear Medicine, especially for evaluation in Region of Interest (ROI) smaller than the Full Width at Half Maximum (FWHM) of the PSF. The aim of this work is to present a new approach for the correction of PVE, using a post-reconstruction process starting from a mathematical expression, which only requires the knowledge of the FWHM of the final PSF of the imaging system used. After the presentation of the theoretical derivation, the experimental evaluation of this method is performed using a PET/CT hybrid system and acquiring the IEC NEMA phantom with six spherical "hot" ROIs (with diameters of 10, 13, 17, 22, 28, and 37 mm) and a homogeneous "colder" background. In order to evaluate the recovery of quantitative data, the effect of statistical noise (different acquisition times), tomographic reconstruction algorithm with and without time-of-flight (TOF) and different signal-to-background activity concentration ratio (3:1 and 10:1) was studied. The application of the corrective method allows recovering the loss of quantification due to PVE for all sizes of spheres acquired, with a final accuracy less than 17%, for lesion dimensions larger than two FWHM and for acquisition times equal to or greater than two minutes.

**Keywords:** partial volume effect; quantitative analysis; point spread function; post-reconstruction-correction-method

## 1. Introduction

Quantitative positron emission tomography/computed tomography (PET/CT) is currently used as a diagnostic/prognostic tool and for assessing therapy efficacy. Quantification in fluorodeoxyglucose (FDG) PET/CT is mainly performed using standardized uptake value (SUV) [1]. Moreover, there is an increasing interest in deriving other quantitative parameters such as the metabolic tumor volume (MTV) and the total lesion glycolysis (TLG) [2].

It is well known that there are several sources of errors in SUV measurements, which are usually even poorly standardized between institutions with different PET equipment. Image reconstruction variability seems to have a prominent role in the unreliability of quantitative assessment of PET images mainly because of improvements in PET technology, which significantly affect SUV measurement. Thus, it has been reported that PET recon-

structions including PSF compensation can increase $SUV_{max}$ more than 66% in small nodal metastases in breast cancer or for NSCLC [3,4].

Moreover, an international survey reported that 52% of PET centers use alternative protocols with adapted reconstruction parameters. Additional complications arise considering the reconstruction variability between centers running similar systems. In fact, it has been reported that site-specific reconstruction parameters increased the quantitative variability among similar scanners [5,6].

The use of post-processing corrective methods could be a way to reduce this variability in order to achieve a better harmonization of SUV measurements between different PET centers.

In this context, quantitative parameters such as the SUV are strictly related to the system spatial resolution performances.

Spatial resolution depends on the physics of PET imaging (positron range, collinearity, scatter), on the detector design, and on the reconstruction method. All these factors lead to some amount of blurring on the images, limiting the final spatial resolution of the system.

The finite spatial resolution affects the quality of the image and the correct estimation of radioactivity concentration, which consequently causes difficulties in the application of a quantitative approach in the evaluation of PET studies [7].

From a theoretical point of view, spatial resolution is defined as the Full Width Half Maximum (FWHM) of the Point Spread Function (PSF) of the imaging system, which is the description of the image response to a point source [8]. Once the PSF is defined for the imaging system, the final image can be modelled as the real radioactivity distribution convolved by the PSF of the imaging system.

The PSF of the PET system is responsible for the so-called partial volume effect (PVE), which affects images both qualitatively and quantitatively [9].

The direct effect of the PSF on the image is due to the fact that the contribution of each ideal point source of radioactivity is spread over a wider volume, i.e., the content of each voxel is the sum of the contributions of neighboring point sources.

This can result in an overestimation of the object size, but also in an underestimation of the real radioactivity concentration, because part of the signal from the source spills out in the background (spill out effect), hence it is seen outside the actual source, as well as, part of the background surrounding the source spreads into the source (spill in effect). The combined effect depends by the source/background activity concentration ratio and by the size of the source. Moreover, from a quantitative point of view, it becomes particularly important when the target is smaller in comparison to the FWHM of the imaging system [10].

In order to achieve an accurate quantification of the radioactivity concentration, PVE has to be considered and compensated, especially in small structures. To date, several techniques have been proposed to compensate for PVE, which are illustrated in details in many papers and reviews [11–13].

The method proposed in this work can be included in the post-reconstruction correction methods applied at regional level. The method is based on the exact analytical deconvolution of the PSF of the imaging system in the case of an activity distribution, which consist of a single source with homogeneous activity, surrounded by a homogeneous background. The method is experimentally tested by means of an IEC NEMA phantom acquisition, and the results are shown and discussed.

## 2. Materials and Methods

### 2.1. Theory

In order to analytically describe how the real counts can be recovered from the image counts, some basic assumptions have been made. Specifically, the method proposed here takes into account only how the system PSF affects the PVE under ideal conditions of uniform radioactive source surrounded by uniform background.

Starting from these assumptions, the post-reconstruction PSF of a 1D PET imaging system can be modeled as a normalized Gaussian function completely described by its standard deviation σ:

$$PSF_{1D} = (x, \sigma) = \frac{1}{\sigma\sqrt{2\pi}} e^{-\frac{x^2}{2\sigma^2}} \qquad (1)$$

In a 1D case, a source uniformly distributed surrounded by a homogeneous background, can be then described by the following function:

$$S_{1D}(x,l) = \begin{cases} s & \forall\, x \in \left[-\frac{l}{2}; \frac{l}{2}\right] \\ b & outside \end{cases} \qquad (2)$$

in which "$s$" is the source counts concentration, "$b$" the background counts concentration, and "$l$" the source dimension. In this case, the image of the source ($L_{1D}$, see Equation (3)) generated by the imaging system characterized by the PSF of Equation (1) is given by the convolution product between the function that describes the source in a uniform background and the PSF, i.e., the convolution between Equations (1) and (2):

$$L_{1D}(x,l,\sigma) = S_{1D}(x, l) \times PSF_{1D}(x,\sigma) \qquad (3)$$

The $S_{1D}(x,l)$ function that describes the source and the background together can be further split into two factors (Equation(4)), related to the source and background, respectively, $S_{1D}(x,l) = S_{1D}^s(x,l) + S_{1D}^b(x)$, where:

$$S_{1D}^s(x,l) = \begin{cases} (s-b) & \forall x \in \left[-\frac{l}{2}; \frac{l}{2}\right] \\ 0 & outside \end{cases} \quad and \quad S_{1D}^b(x) = b\forall x \in [-\infty; \infty] \qquad (4)$$

By considering the linearity properties of the convolution operator, it follows that:

$$\begin{aligned} L_{1D}(x,l,\sigma) &= \left[S_{1D}^s(x,l) + S_{1D}^b(x)\right] \times PSF_{1D}(x,l) = \\ & S_{1D}^s(x,l) \times PSF_{1D}(x,\sigma) + S_{1D}^b(x) \times PSF_{1D}(x,\sigma) = \\ & (s-b)\frac{1}{\sigma\sqrt{2\pi}} \int_{-\frac{l}{2}}^{\frac{l}{2}} e^{-\frac{(x-x')^2}{2\sigma^2}} dx' + b\frac{1}{\sigma\sqrt{2\pi}} \int_{-\infty}^{+\infty} e^{-\frac{(x-x')^2}{2\sigma^2}} dx' \\ & (s-b)I_{1D}(x,l,\sigma) + b \end{aligned} \qquad (5)$$

in which the first integral of Equation (5) can be seen as the sum of two error functions (erf) and it is indicated with $I_{1D}(x,l,\sigma)$ (Equation (6)), while the second one results 1. Therefore, the $I_{1D}(x,l,\sigma)$ function can be written as:

$$I_{1D}(x,l,\sigma) = \frac{1}{2}\left[erf\left(\frac{x+\frac{l}{2}}{\sigma\sqrt{2}}\right) - erf\left(\frac{x-\frac{l}{2}}{\sigma\sqrt{2}}\right)\right] \qquad (6)$$

The average value of the counts concentration in the image ($C_{image}$) can then be obtained by integrating Equation (3), within the source bounds $[-l/2; +l/2]$ and dividing by l:

$$\begin{aligned} C_{image} &= \frac{1}{l} \int_{\frac{-l}{2}}^{\frac{l}{2}} L_{1D}(x)dx = (s-b)H_{1D}(l,\sigma) + b = \\ & sH_{1D}(l,\sigma) + b(1 - H_{1D}(l,\sigma)) \end{aligned} \qquad (7)$$

where $H_{1D}(l,\sigma)$ in Equation (7) describes the effect of the PSF on the image counts and can be written as:

$$H_{1D}(l,\sigma) = \frac{1}{l} \int_{\frac{-l}{2}}^{\frac{l}{2}} L_{1D}(x)dx = \frac{\int_{\frac{-l}{2}}^{\frac{l}{2}} I_{1D}(x, l, \sigma)dx}{\int_{\frac{-l}{2}}^{\frac{l}{2}} I_{1D}(x, l, \sigma)dx} = \frac{\sigma}{l}\frac{\sqrt{2}}{\sqrt{\pi}}\left(e^{\frac{-l^2}{2\sigma^2}} - 1\right) + erf\left(\frac{\frac{l}{2}}{\sigma\sqrt{2}}\right) \qquad (8)$$

From Equation (7) and Equation (8) it is possible to obtain the value of the source counts concentration s (see Equation (9)), as a function of the background counts concentration b, $C_{image}$ and $H_{1D}(l,\sigma)$:

$$s = \frac{1}{H_{1D}(l,\sigma)}C_{image} - b\frac{1 - H_{1D}(l,\sigma)}{H_{1D}(l,\sigma)} \tag{9}$$

Equation (9) shows that it is possible to quantify s from the knowledge of:

(1) the function $H_{1D}(l,\sigma)$, which is an analytic function of measurable parameters: the dimension of the source and the $\sigma$ of the image system's PSF;

(2) $C_{image}$ directly measurable on the image space;

(3) *b* that, in the practice, can be evaluated considering a background region of interest (ROI) away from the source (i.e., away with respect to the FWHM of the system's PSF) on the image space.

It is important to notice that:

(1) $0 \leq H_{1D}(l,\sigma) \leq 1$;

(2) $\frac{1}{H_{1D}(l,\sigma)}$ is the factor which corrects the effect of loss of counts on the image space inside the ROI (*spill out*);

(3) $\frac{1 - H_{1D}(l,\sigma)}{H_{1D}(l,\sigma)}$ is the factor that corrects the effect of increase of counts on the image space inside the ROI due to the background (*spill in*).

The proposed approach can now be generalized in a 3D ideal scenario (Equation (10)), i.e., by considering a 3D source surrounded by a homogeneous background, which can be described as:

$$S_{3D} = (x,l_x;y,\,l_y;z,l_z) = \begin{cases} s & \forall(x,y,z) \in \left[\frac{-l_x}{2};\frac{l_x}{2}\right];\ \left[\frac{-l_y}{2};\frac{l_y}{2}\right];\ \left[\frac{-l_z}{2};\frac{l_z}{2}\right] \\ b & outside \end{cases} \tag{10}$$

Since the PSF of the 3-D (three-dimensional) imaging system can be derived as:

$$PSF_{3D}(x,\,\sigma_x;y,\sigma_y;z,\,\sigma_z) = \frac{1}{(\sigma_x,\sigma_y\sigma_z)\sqrt{8\pi^3}}e^{-\frac{x^2}{\sigma_x^2}-\frac{y^2}{\sigma_y^2}-\frac{z^2}{\sigma_z^2}} = \tag{11}$$
$$PSF_{1D}(x,\,\sigma_x)PSF_{1D}(x,\,\sigma_y)PSF_{1D}(x,\,\sigma_z)$$

it is easy to verify that:

$$I_{3D}(x,l_x,\sigma_x;y,l_y,\sigma_y;z,l_z,\sigma_z) = \frac{1}{(\sigma_x,\sigma_y\sigma_z)\sqrt{8\pi^3}}\int_{\frac{-l_x}{2}}^{\frac{l_x}{2}}\int_{\frac{-l_y}{2}}^{\frac{l_y}{2}}\int_{\frac{-l_z}{2}}^{\frac{l_z}{2}}e^{-\frac{(x-x')^2}{\sigma_x^2}-\frac{(y-y')^2}{\sigma_y^2}-\frac{(z-z')^2}{\sigma_z^2}}\,dx'dy'\,dz' = \tag{12}$$
$$I_{1D}(x,l_x,\sigma_x) \times I_{1D}(y,l_y,\sigma_y) \times I_{1D}(z,l_z,\sigma_z)$$

and

$$H_{3D}(x,\,\sigma_x;y,\,\sigma_y;z,\,\sigma_z) = \frac{\int_{\frac{-l_x}{2}}^{\frac{l_x}{2}}\int_{\frac{-l_y}{2}}^{\frac{l_y}{2}}\int_{\frac{-l_z}{2}}^{\frac{l_z}{2}}I_{3D(x,l_x,\,\sigma_x;y,l_y,\,\sigma_y;z,l_z,\,\sigma_z)}dxdydz}{\int_{-\infty}^{\infty}\int_{-\infty}^{\infty}\int_{-\infty}^{\infty}I_{3D(x,l_x,\,\sigma_x;y,l_y,\,\sigma_y;z,l_z,\,\sigma_z)}dxdydz} = \tag{13}$$
$$H_{1D}(l_x,\sigma_x)H_{1D}(l_y,\sigma)H_{1D}(l_z,\sigma_z)$$

From Equations (12) and (13), it is possible to derive Equation (14), which describes the final 3D generalization of Equation (7) simply as:

$$s = \frac{1}{H_{1D}(l_x,\sigma_x)H_{1D}(l_y,\sigma_y)H_{1D}(l_z,\sigma_z)}C_{image} - b\frac{1 - H_{1D}(l_x,\sigma_x)H_{1D}(l_y,\sigma_y)H_{1D}(l_z,\sigma_z)}{H_{1D}(l_x,\sigma_x)H_{1D}(l_y,\sigma_y)H_{1D}(l_z,\sigma_z)} \tag{14}$$

## 2.2. Scanner and Phantom

A Discovery 710 PET/TC hybrid system (General Electric Company, Boston, MA, USA) [14] was employed for images acquisition. The PET system collects data in three-dimensional (3D) mode and can reconstruct images with or without time-of-flight technol-

ogy (TOF) [15]. The CT system allows performing data attenuation correction and provides morphological information through automatic image co-registration [16].

A standard NEMA IEC body phantom was adopted in this study [17,18]. The phantom mimics the D-shape of an upper human body; it comprises a cylindrical insert in the center of phantom and 6 fillable spheres of different sizes. The spheres are suspended in the D-shaped cavity with the six centers placed on the same plane and with nominal inner diameters of 10, 13, 17, 22, 28, and 37 mm. The nominal volume of the phantom is 9.7 L ($\pm$1%), excluding the volume of the six spheres.

### 2.3. Experimental Measurements

Two sets of acquisitions were performed in order to apply our correction model. First, point source images were acquired to estimate the point spread function (PSF) of the system [19,20] under different conditions, then, a series of acquisitions of the IEC phantom under the same conditions were performed to test the recovery method.

### 2.4. Point Source Measurements

An 18 F-FDG point-like source was prepared with an activity of 0.1 mCi. The point-like source was simulated using the needle of a syringe placed in air through a mechanical support.

The system PSF was evaluated under different conditions in order to match the IEC NEMA phantom acquisitions. The measurements were performed by placing the source at the center of the field of view (FOV) and a 5 cm off-center, in order to highlight any anisotropies of the acquisition system.

The acquisitions were corrected for the attenuation, scatter, radionuclide decay, dead time of the detectors, random coincidences, and detector normalization; 47 slices were reconstructed with a matrix size of $256 \times 256$; the corresponding voxel sizes were $2.73 \times 2.73 \times 3.27$ mm$^3$. The acquisition and reconstruction parameters were the same as the corresponding IEC phantom measurements described in the following paragraphs.

### 2.5. IEC Phantom Measurements

The volume of the NEMA phantom and the six spheres were filled with $^{18}$F-FDG mixed with pure water using two different signal-to-background (SBR) activity concentration ratio (around 3:1 and 10:1). In order to obtain the concentration on the spheres and background with accuracy, the following procedure was established: first, the initial activity was measured through an activity calibrator (Atomlab 500 Dose Calibrator, Biodex, New York, NY, USA); then, the concentration of the sphere was obtained by mixing the measured activity with pure water and the spheres were filled through a precision syringe; finally, the phantom (background) was filled with the remaining 18 F-FDG solution by adding pure water in order to obtain the 3:1 and 10:1 SBRs. Special care was adopted in order to ensure a proper homogeneity of the radioactive solution.

The final concentrations are reported in Table 1. Once the phantom was set up and an accurate alignment with the FOV center achieved, a series of acquisitions were performed in order to study the effect of different acquisition/reconstruction conditions on the quantitative recovery model.

**Table 1.** This table reports the two final signal-to-background with the respective concentrations in spheres and in background.

| SBR | Spheres (kBq/mL) | Phantom Background (kBq/mL) |
|---|---|---|
| 3.27 | 17.27 | 5.28 |
| 10.18 | 35.94 | 2.55 |

The main acquisition and reconstruction parameters are reported in Table 2. All the data were corrected for the attenuation, scatter, radionuclide decay, dead time of the detectors, random coincidences, and detector normalization. The complete details

of the image reconstruction process are reported in Table 3. Both PET and CT images were collected in DICOM (Digital Imaging and Communications in Medicine) format and subsequently analyzed.

**Table 2.** Acquisition parameters for the phantom acquisitions.

| Acquisition Time (min) | Image Matrix Size | Pixel Dimension (mm$^2$) | Slice Thickness (mm) | Number of Reconstructed Slices |
|---|---|---|---|---|
| 1, 2, 4, 8 | 256 × 256 | 2.73 × 2.73 | 3.27 | 47 |

**Table 3.** Reconstruction parameters of the algorithm.

| | Reconstruction Algorithm | Iteration Subs | Gaussian Filter | Reconstruction Enhancement |
|---|---|---|---|---|
| PET TOF | VUE Point FX | 3 iteration 24 subs | 5.5 mm | SharpIR |
| PET noTOF | VUE Point HD | 3 iteration 24 subs | 5.5 mm | SharpIR |

### 2.6. Data Analysis

Images and data analysis were performed by means of in-house scripts running on Matlab (The MathWorks, Inc., Natick, MA, USA) software package.

### 2.7. Point Spread Function Assessment

Images of the point source were employed to estimate the PSF of the system, which was characterized by the standard deviation (σ), assuming a Gaussian profile [7]. The three horizontal (x), vertical (y), and axial (z) profiles were extracted from the central region of the image and a Gaussian fit was performed.

### 2.8. Segmentation Method

In order to study how the PVE quantitatively affects the image, the first step was to determine the number of the counts ($C_{image}$) inside each of the six spheres of the phantom in the original image. Since the PVE alters the activity distribution making the contours of the volume of interest (VOI) undefined, an ideal segmentation criterion was implemented, using the geometrical information from the CT acquisition.

This criterion consisted of constructing a binary mask of six spherical VOIs with radii R equal to the phantom spheres and centered on the x and y coordinates of their centers, determined from the CT acquisition.

The discretization of the sphere brought to the problem that some voxels on the boundary were partially included in the spherical VOIs. To overcome that, an inclusion criteria has been chosen in such a way that only the voxels which have all the corners with a distance from the coordinates of center less than the radius were considered inside.

These voxel completely included in the VOI have a unitary value, while a value of 0 is assigned to the voxels outside the VOI.

In order to improve the accuracy of the segmentation, the image matrix of PET acquisition were first resampled on a finer grid, i.e., each voxel was divided in smaller voxels. The effect of the resampling on the counts was investigated, starting from the original matrix (256 × 256 × 47) until a 1536 × 1536 × 282 matrix with unitary step.

A voxel by voxel product between the PET image matrix and the binary mask has been calculated in order to select only the counts of the voxels corresponding to the interior of the VOIs.

*2.9. Counts Recovery*

Once the PSF of the system was characterized and the counts for the original image ($C_{image}$) in the six spheres of the phantom images obtained, the recovery formula (Equation (14)) was applied in order to obtain the resultant counts ($C_{recover}$) and assess the accuracy of the PVE correction method under different conditions.

The recovery after applying PVE correction, estimated as percentage difference between $C_{image}$ and $C_{recover}$ was estimated for each comparison.

## 3. Results

With the purpose of evaluating the efficiency in PVE recovery, the method was applied changing one parameter at a time:

1.  to investigate the effect of quantum noise both on PVE and on the recovery: fixed SBR and reconstruction algorithm, different acquisition time (1, 2, 4, and 8 min);
2.  to investigate the effect of the reconstruction algorithm on the system PSF and PVE: fixed SBR and same acquisition time (4 min), different reconstruction algorithm with TOF (VPFXS) and without TOF (VPHDS) [15];
3.  to investigate the effect of different source-to-background concentration ratio on PVE: same acquisition time (4 min) and reconstruction algorithm, different SBRs (3:1 and 10:1).

*3.1. PSF*

The point-like source was acquired for all the different configurations above, collecting the σ needed for the analysis that follows. The off-center measurements did not show appreciable variations, hence we consider the σ values obtained at the center of the FOV.

Figure 1 reports an example for the 2 min acquisition and reconstruction method with TOF correction. Each of the mono-dimensional data sets were fitted with a Gaussian function, from which the standard deviations were obtained with confidence interval set at the 95% level. The values were respectively $2.54 \pm 0.01$ mm, $2.76 \pm 0.03$ mm, and $3.31 \pm 0.04$ mm.

*3.2. Resampling*

The effect of resampling on the counts within the spheres is illustrated in Figure 2. The x-axis reports the resampling factor where a factor of 1 represents the original image matrix ($256 \times 256 \times 47$), while the maximum resampling resulted in a $1792 \times 1792 \times 329$ matrix with a corresponding factor of 7; the y-axis reports the counts ratio between the counts obtained in the resampled image and the counts associated to the original image.

The accuracy in gain can be appreciated especially in the smallest sphere ("sphere 6" blue dots in Figure 2), where the counts on the resampled image are three times the original ones.

As can be seen in Figure 2, the counts ratio increases with increasing the resampling factor with an asymptotic behavior, which has been verified by fitting the data with the function $f(x) = a - b \times c^x$ and studying the horizontal asymptote (i.e., the *a* parameter). Table 4 reports the asymptotes derived from the fit (with confidence level set at 95%) and the count ratio obtained with a resampling factor of six (i.e., the factor adopted in the following analysis, which corresponds to an image matrix of $1536 \times 1536 \times 282$). The counts ratios associated to a resampling factor of six were very close to the asymptote *a* obtained from the fit. Moreover, a finer matrix of $1536 \times 1536 \times 282$ respect to the original one represents a good compromise between the accuracy in the counts recovery and the increase in computation time. The increment from a resampling factor of six to seven was below the 1.5%, but the computation time became three times higher.

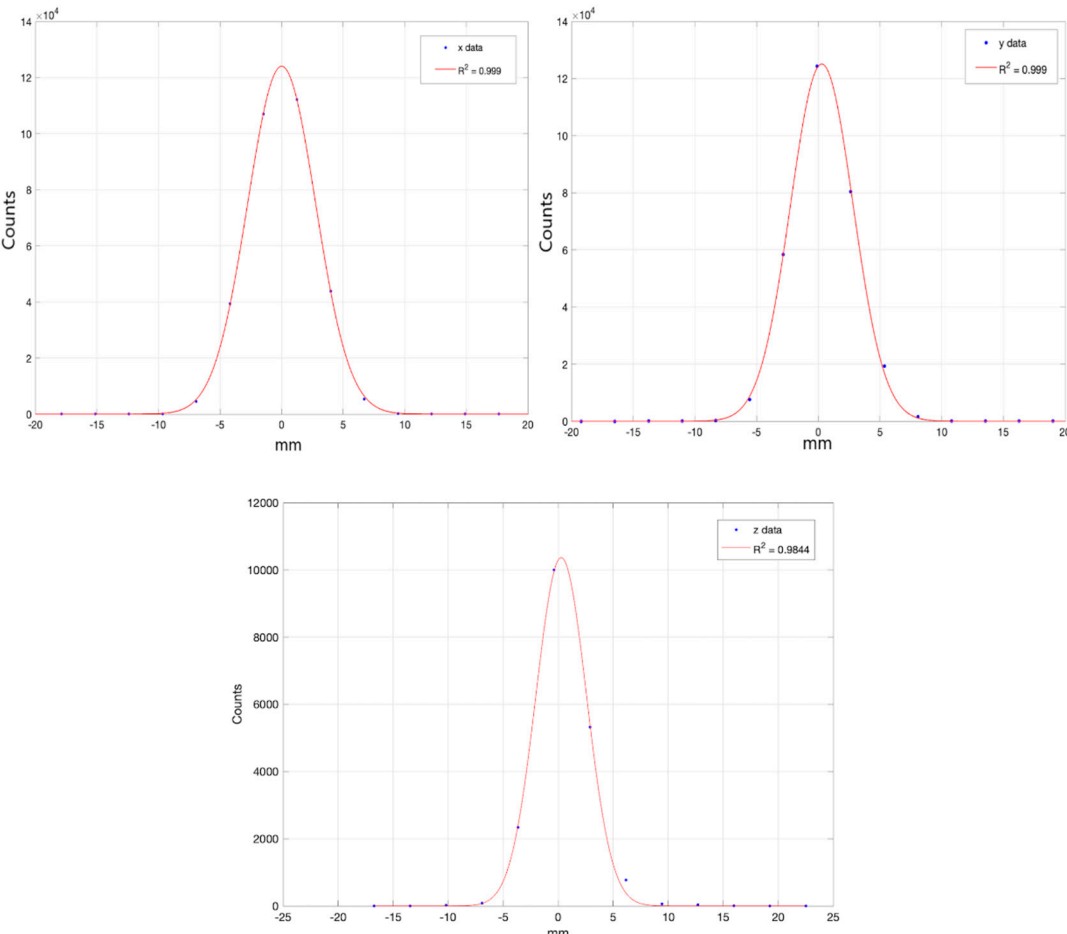

**Figure 1.** Horizontal (x), vertical (y), and axial (z) profiles extracted from the point-like source fitted with a Gaussian function.

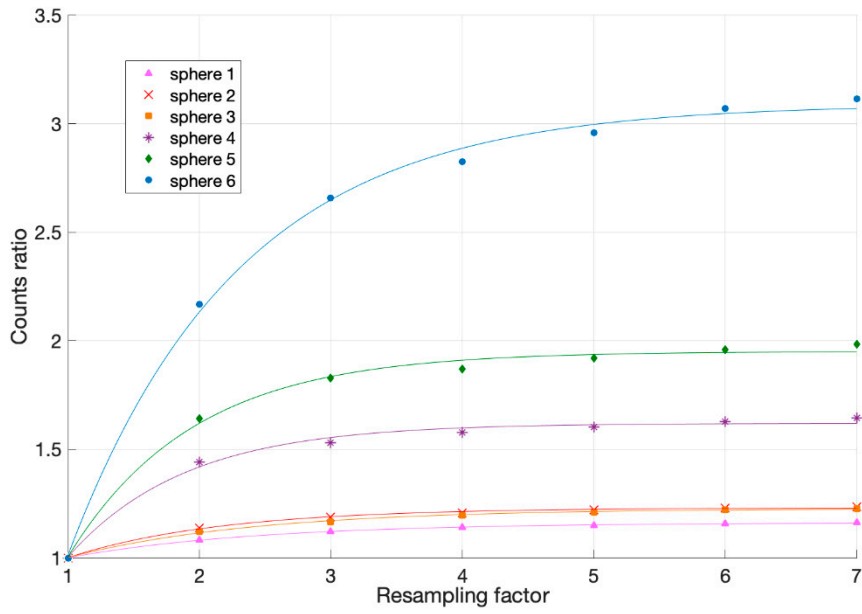

**Figure 2.** Counts ratio normalized to the original counts at the increasing of the voxel division of the starting matrix ($256 \times 256 \times 47$) for each sphere, marked in different color with different markers.

**Table 4.** Parameters obtained by the fit with the function $f(x) = a - b \times c^x$. The $a$ parameter represents the asymptote.

| Spheres | 1 | 2 | 3 | 4 | 5 | 6 |
|---|---|---|---|---|---|---|
| $a$ | $1.16 \pm 0.04$ | $1.23 \pm 0.01$ | $1.23 \pm 0.01$ | $1.62 \pm 0.04$ | $1.95 \pm 0.04$ | $3.09 \pm 0.09$ |
| *Counts ratio for a resampling factor of 6* | $1.16 \pm 0.01$ | $1.23 \pm 0.01$ | $1.22 \pm 0.01$ | $1.63 \pm 0.01$ | $1.96 \pm 0.01$ | $3.07 \pm 0.02$ |
| $b$ | $0.32 \pm 0.03$ | $0.54 \pm 0.07$ | $0.45 \pm 0.06$ | $1.90 \pm 0.70$ | $2.70 \pm 0.80$ | $4.50 \pm 0.70$ |
| $c$ | $0.50 \pm 0.03$ | $0.42 \pm 0.07$ | $0.50 \pm 0.07$ | $0.30 \pm 0.10$ | $0.30 \pm 0.10$ | $0.46 \pm 0.07$ |

### 3.3. Evaluation of PVE Recovery

Figures 3–5 show the sphere-to-background counts concentration ratio as a function of the spheres dimension for the different situations above. However, in order to underline how the relationship between the dimension of lesions and the PSF of the imaging system impacts on PVE, the x-axes are reported as the ratio between the diameter of the six spheres and the σ of the system.

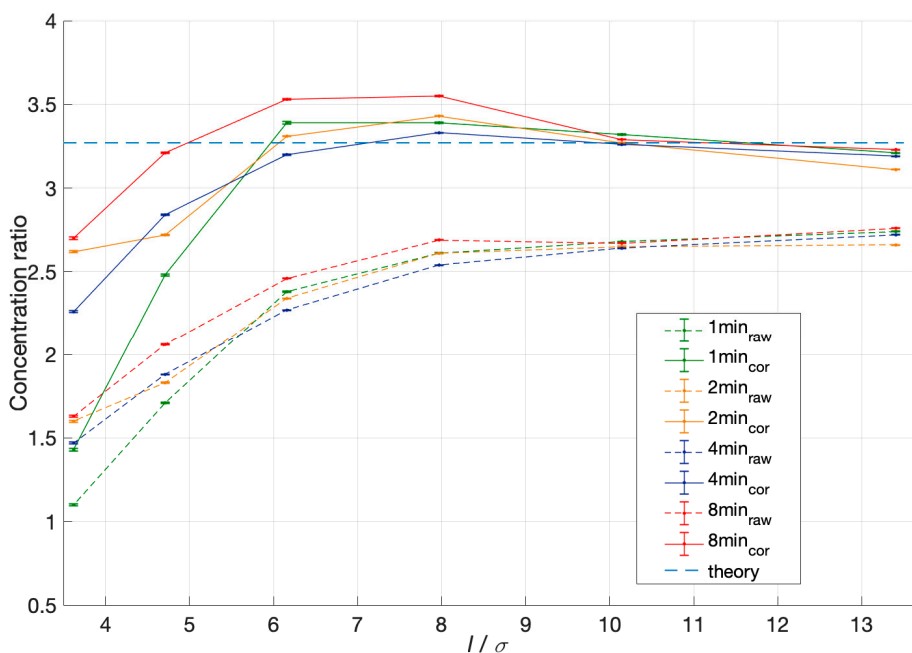

**Figure 3.** Comparison between the concentration ratio for the raw image (dotted lines) and the PVE corrected image (continuous line) at different acquisition times, shown with different colors.

The sphere-to-background counts concentration ratio was calculated as the ratio between the counts concentration in the spheres and in the background, which were measured as the average over four circular ROIs chosen in areas and slices far from the spheres.

Figure 3 illustrates the comparison in concentration ratio (y-axis) before and after the application of the PVE recovery method at different acquisition times (1, 2, 4, and 8 min).

The SBR and the reconstruction algorithm VPFXS were fixed.

For each acquisition time, reported with the same colors, the smaller the lesion, the greater results the underestimation of the concentration ratio. In particular, the concentration underestimation ranges from a minimum of 16% for largest spheres to a maximum of 66% for the smallest sphere.

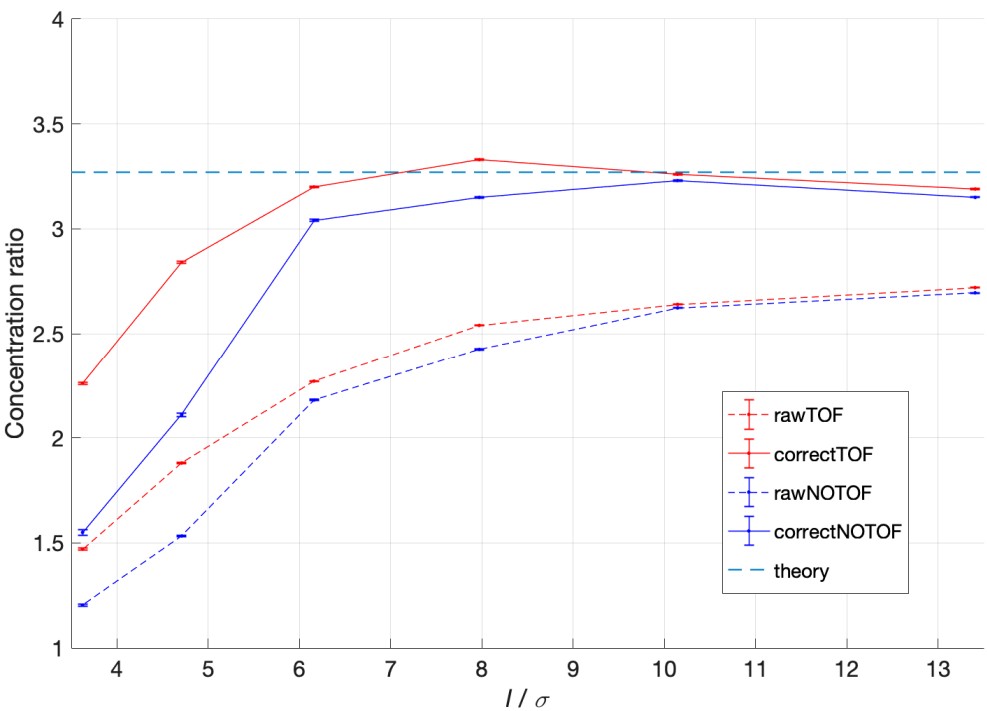

**Figure 4.** Comparison of the concentration ratio for the same acquisition time (4 min) between the reconstruction with TOF correction, reported in red, and the one without TOF, reported in blue, before (dotted line) and after (continuous line) the application of PVE recovery method.

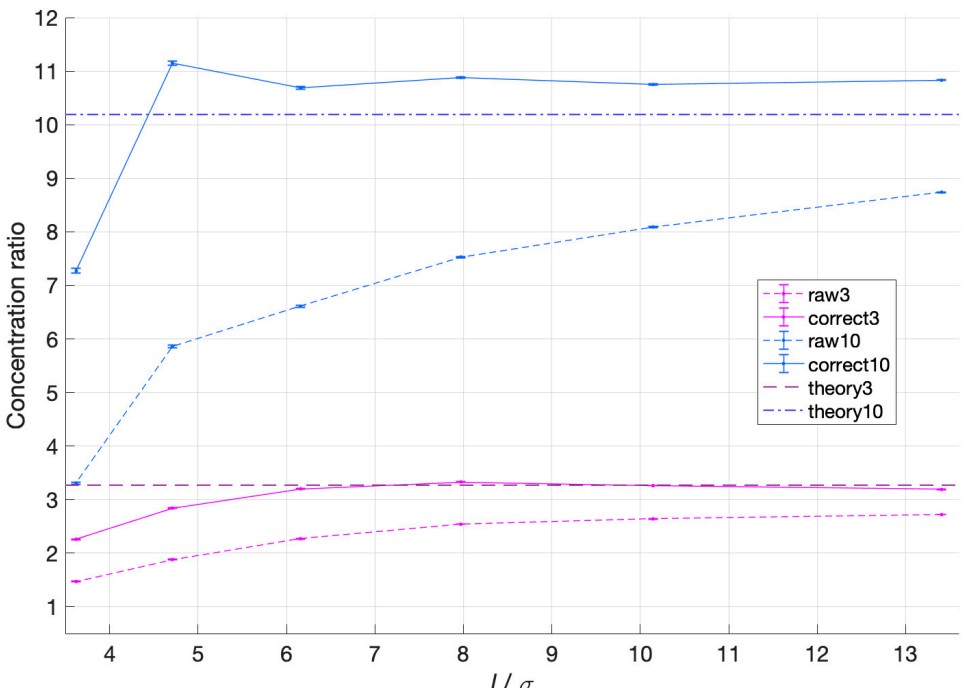

**Figure 5.** Comparison for concentration ratio for SBR = 10.18 in light blue and SBR = 3.27 in violet. For both acquisitions, acquisition time was fixed at 4 min and the reconstruction algorithm was with TOF correction. The dotted lines represent the raw images, while the continuous ones the PVE corrected images.

Table 5 shows the percentage of recovery for all the spheres of the NEMA phantom for all the acquisition time, highlighting the good PVE recovery for the biggest sphere, which for all the acquisition time results in the range of ±3%.

**Table 5.** Comparison of percentage of recovery at different acquisition time for the six spheres of NEMA phantom, where the dimensions are expressed as the ratio between the diameter of the six spheres and the PSF of the system.

| $l/\sigma$ | 3.6 | 4.7 | 6.2 | 8.0 | 10.1 | 13.4 |
|---|---|---|---|---|---|---|
| $1\ min_{raw}$ | $-66 \pm 2\%$ | $-48 \pm 2\%$ | $-27 \pm 2\%$ | $-20 \pm 1\%$ | $-17.8 \pm 0.8\%$ | $-16.3 \pm 0.5\%$ |
| $1\ min_{cor}$ | $-56 \pm 2\%$ | $-24 \pm 2\%$ | $+4 \pm 2\%$ | $+3 \pm 1\%$ | $+1.4 \pm 0.9\%$ | $-1.9 \pm 0.6\%$ |
| $2\ min_{raw}$ | $-51 \pm 2\%$ | $-44 \pm 1\%$ | $-28.6 \pm 0.9\%$ | $-20.3 \pm 0.7\%$ | $-19.0 \pm 0.6\%$ | $-17.6 \pm 0.5\%$ |
| $2\ min_{cor}$ | $-20 \pm 2\%$ | $-17 \pm 1\%$ | $+1 \pm 1\%$ | $+4.9 \pm 0.8\%$ | $+0.0 \pm 0.7\%$ | $+3.5 \pm 0.5\%$ |
| $4\ min_{raw}$ | $-55 \pm 2\%$ | $-42 \pm 1\%$ | $-30.5 \pm 0.9\%$ | $-22.2 \pm 0.7\%$ | $-19.3 \pm 0.6\%$ | $-16.7 \pm 0.5\%$ |
| $4\ min_{cor}$ | $-30 \pm 2\%$ | $-13 \pm 1\%$ | $-2 \pm 1\%$ | $+1.0 \pm 0.8\%$ | $+0.4 \pm 0.7\%$ | $-2.5 \pm 0.6\%$ |
| $8\ min_{raw}$ | $-50 \pm 2\%$ | $-37 \pm 1\%$ | $-24.7 \pm 0.9\%$ | $-17.9 \pm 0.7\%$ | $-18.4 \pm 0.6\%$ | $-15.7 \pm 0.5\%$ |
| $8\ min_{cor}$ | $-17 \pm 2\%$ | $-1.9 \pm 1\%$ | $+7\% \pm 1\%$ | $+8.3 \pm 0.9\%$ | $+0.7 \pm 0.7\%$ | $-1.2 \pm 0.6\%$ |

Figure 4 compares the concentration ratio with or without the TOF reconstruction for the 4 min acquisition, showing that every reconstruction underestimates the theoretical SBR, even if the TOF correction tends to have a proximity of 5% more than the NOTOF.

After the recovery method application (continuous line), the PVE compensation is shown in Table 6.

**Table 6.** Comparison of percentage of recovery for the same acquisition time (4 min) between the reconstruction with TOF correction and the one without TOF.

| $l/\sigma$ | 3.6 | 4.7 | 6.2 | 8.0 | 10.1 | 13.4 |
|---|---|---|---|---|---|---|
| $TOF_{raw}$ | $-55 \pm 2\%$ | $-42 \pm 1\%$ | $-30.5 \pm 0.9\%$ | $-22.2 \pm 0.7\%$ | $-19.3 \pm 0.6\%$ | $-16.7 \pm 0.5\%$ |
| $TOF_{cor}$ | $-30 \pm 2\%$ | $-13 \pm 1\%$ | $-2 \pm 1\%$ | $+1.0 \pm 0.8\%$ | $+0.4 \pm 0.7\%$ | $-2.5 \pm 0.6\%$ |
| $NOTOF_{raw}$ | $-55 \pm 2\%$ | $-45 \pm 1\%$ | $-33.3 \pm 0.9\%$ | $-26.5 \pm 0.7\%$ | $-21.4 \pm 0.6\%$ | $-20.1 \pm 0.5\%$ |
| $NOTOF_{cor}$ | $-31 \pm 2\%$ | $-19 \pm 1\%$ | $+7 \pm 1\%$ | $+4.6 \pm 0.8\%$ | $+3.3 \pm 0.7\%$ | $+6.8 \pm 0.6\%$ |

Finally, Figure 5 illustrates the change in SBRs with the theoretical concentration ratio of 3.27 and 10.19, maintaining unaltered the other parameters (acquisition time = 4 min and TOF reconstruction).

For higher SBR, the underestimation of theoretical ratio concentration results almost always greater, as shown in Table 7.

**Table 7.** Comparison of percentage of recovery for the different SBR. For both acquisitions, acquisition time was fixed at 4 min and the reconstruction algorithm was with TOF correction.

| $l/\sigma$ | 3.6 | 4.7 | 6.2 | 8.0 | 10.1 | 13.4 |
|---|---|---|---|---|---|---|
| $SBR3_{raw}$ | $-55 \pm 2\%$ | $-42 \pm 1\%$ | $-30.5 \pm 0.9\%$ | $-22.2 \pm 0.7\%$ | $-19.3 \pm 0.6\%$ | $-16.7 \pm 0.5\%$ |
| $SBR3_{cor}$ | $-30 \pm 2\%$ | $-13 \pm 1\%$ | $-2 \pm 1\%$ | $+1.0 \pm 0.2\%$ | $+0.4 \pm 0.7\%$ | $-2.5 \pm 0.6\%$ |
| $SBR10_{raw}$ | $-67.5 \pm 0.6\%$ | $-42.5 \pm 1\%$ | $-35.1 \pm 0.2\%$ | $-26.1 \pm 0.2\%$ | $-20.6 \pm 0.1\%$ | $-14.21 \pm 0.07\%$ |
| $SBR10_{cor}$ | $-28.7 \pm 0.7\%$ | $+9.4 \pm 1\%$ | $+4.9 \pm 0.3\%$ | $+6.7 \pm 0.2\%$ | $+5.5 \pm 0.1\%$ | $+6.27 \pm 0.07\%$ |

## 4. Discussion

In this work, we developed a post-reconstruction method for the correction of the PVE in PET images. Quantitative PET images are increasingly required for diagnostic

(SUV) and therapeutic (committed dose in metabolic radiotherapy) purposes and radiomics applications [21]. In general, post-processing corrective methods offer multiple advantages with respect to those integrated into the reconstruction algorithm [19]:

(1)     they do not need access to the reconstruction-code, which is practically impossible for the standard user, and they can be easily used;

(2)     they allow their application independently of the acquisition machine and, consequently, can be used to standardize data from different sites, for example for multi-center studies.

Our method is based on the theoretical calculation of the PVE effect caused by a uniformly distributed source on a uniform background. The hypotheses on which it is "exactly" valid are:

(1)     uniform lesion on a uniform background

(2)     ideal segmentation of the ROI (Region Of Interest), or by means of CT imaging

Its application is very simple, as it only needs:

(1)     to measure the contribution of the background directly on the image;

(2)     to know the PSF (Point Spread Function) of the acquisition system, acquiring a linear or point source or taking the data directly from the NEMA post installation quality assurance;

(3)     to implement a simple analytical formula in a spreadsheet.

The limitations and the field of application of the method are identical to those present in the well-known and used RC (Recovery Coefficient) method [22], but, unlike the latter, it has the great advantage that the final formula has been obtained theoretically; therefore, the final formula intrinsically contains the dependencies of the various parameters that influence the PVE: size of the lesion, lesion/background concentration ratio, and FWHM of the PSF of the system. For this reason, it is not necessary, as is the case for the RC method, to measure experimentally these parameters every time they change.

The method was experimentally validated by acquiring an IEC NEMA PHANTOM, which allowed to evaluate the recovery of the quantitative data by varying:

(1)     the size of the lesions, filling spheres with a diameter of 10, 13, 17, 22, 28, and 37 mm.

(2)     the signal-to-background activity concentration ratio (SBR); in particular we have acquired the values of around 3:1 and 10:1, which covers the standard range of clinical variation in terms of the lesion/background concentration ratio.

(3)     the statistical noise, acquiring the same activity for different acquisition times (1, 2, 4, and 8 min)

(4)     the tomographic reconstruction algorithm, with and without TOF.

The method guarantees an excellent recovery of the quantitative data in the analyzed conditions. Considering the sigma that characterizes the acquisition system, which is around 2.5 mm (value measured experimentally), the PVE effect determines a loss of counts greater than 50% and the corrective method allows to obtain accuracies of less than 17%, for lesion dimensions equal to or greater than 13 mm in diameter and for acquisition times equal to or greater than 2 min. By increasing the acquisition time, substantial improvements are not appreciated and this is an important and comfortable result, given that an acquisition time of 2 min per bed is the standard clinical PET acquisition time.

Additionally, the application of TOF during the reconstruction algorithm inherently reduces the effect of PVE and the application of the corrective method at this data allows obtaining results that are even more accurate.

Finally, increasing the SBR, in general makes the effect of PVE to be more marked and the accuracy of the corrected data is lower.

## 5. Conclusions

The proposed approach for the correction of the PVE is an alternative post-processing method to the RC method; it has the same limitations, but the advantage of having

theoretically included the dependencies of the parameters that describe the effect and therefore to overcome the need of measuring these parameters every time they change

It represent a straightforward method to implement and adopt on every PET system. Although its field of application is widespread, its limits are related to the fact that the method is based on an "ideal" segmentation and does not take into account any inhomogeneities within the ROI considered; both of these aspects can be addressed using the same formalism proposed in the theoretical part of Materials and Methods, and are the topics of future works we are working on.

**Author Contributions:** Conceptualization, F.D.M.; data curation, F.D.M., P.B., E.B., A.G., D.V.; writing—original draft preparation, P.B., E.B., A.G.; writing—review and editing, F.D.M., P.B., E.B., A.G., D.V.; supervision, F.D.M., D.V.; funding acquisition, D.V. All authors have read and agreed to the published version of the manuscript.

**Funding:** This research received no external funding.

**Institutional Review Board Statement:** Not applicable.

**Informed Consent Statement:** Not applicable.

**Conflicts of Interest:** The authors declare no conflict of interest.

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
