# Peer review of "Correction for the Partial Volume Effects (PVE) in Nuclear Medicine Imaging: A Post-Reconstruction Analytic Method"

_applsci, doi:10.3390/app11146460_

Round 1

Reviewer 1 Report

This paper represents a solution to the issue that arises during determination of the activity obtained during nuclear medicine measurements whereby the partial volume effect can lead to incorrect activity measurements.

This paper discusses how these effects occur and develops a simple theoretical model to describe them and correct for them in a clear and easy-to-understand way.

Following the formulation of this correction, the method is then applied to experimental data obtained using phantoms where a careful investigation of the confounding variables has been performed.

The technique can be seen to improve the activity measurement obtained for a variety of situations where the partial volume effect can clearly be seen in their data.

Overall, this paper explains very clearly the problem and solution in a scientifically sound way. I believe the paper should be accepted be publication following these corrections:

Overall comments:

-Figures need larger font sizes to be more clearly read

-There appear to be issues with the formula formatting, please attempt to rectify this

Specific line corrections:
-L19 (ROI) small - missing words - "that is small"?
-L35 "2 minutes with an activity of XX"
-L38 Define FDG
-L206 point-like
-L207 simulateD using THE needle of a
-L216 mm^3 needs to be superscript
-L216 were the same as THE corresponding
-L220 18F superscript
-L241 DICOM - add reference or explain acronym
-L250,252,316 Gaussian to be capitalized
-L253 quantitivatively affects (re-order)
-L260 This criteriON consisted OF
-L266 where->were
-L300 point-like
-L302 each of the mono-dimensional data sets
-L303 missing word? ", and were"
-L304 confidence interval set at the 95% level
-L314 Figure for Gaussian profiles needs to have much larger axis labels, very hard to read!
-L315 Figure caption missing figure number
-L355 e --> and
-L360, "the smaller the lesion, the greater the underestimation of the concentration ratio"
-L368 capital S at end of dimensionS
-L373 Figure 4 caption is split around the figure 
-L379 (continuous line), the PVE compensation is shown in Table 5
-L392 concentration results IS almost always greater

Author Response

Thank you for for evaluating and reviewing our manuscript, we are especially grateful for commenting on the value of our work.

Figures need larger font sizes to be more clearly read.

We modified the font in the figures.

There appear to be issues with the formula formatting, please attempt to rectify this.

We had problems changing the Word format. We fixed it in the revised article.

Specific line corrections

All spelling, grammatical and specif line corrections pointed out have been corrected.  They were helpful for improving the manuscript.

Thank you again.

Reviewer 2 Report

This work shows a practical attempt to consider a simplistic alternative solution to the PVE which limit true quantification in PET.

This work is at present marred by several areas that need attention.

Firstly ;Many of the important details are missing, such as the exact details of the reconstruction used (Iterations, subsets ,corrections etc)

Secondly , there is no attempt to compare the results with that of a conventional Resolution recovery algorithm within the reconstruction.  So no advantage (or not) is demonstrated.

Question,  was this the optimum reconstruction protocol, or could you prove that this form of PVE correction  could be applied to any reconstruction (supposing PSFs could be derived)  It is all linked back to the reconstruction used, which was not made clear.  What reconstructions were available  Sharp IR, or possible QClear?

Particularly references to up to date comprehensive studies such as:-

a)Experimental validation of estimated spatially variant radioisotope specific point spread functions using published positron range simulations and fluorine-18 measurements.  Phys. Med. Biol. 63 (2018) 24           or 

b) Point-spread function reconstructed PET images of sub-centimeter lesions are not quantitative EJNMMI Phys. 2017 Dec; 4: 5.
would help with arguing if corrections based on a simple single Gaussian assumption is enough. 
What was the variation in PSF when measured off centre?  Was this included in any of the correction used?
What level of error would be acceptable with this technique? Compared to other techniques is it more robust over a range of volumes?
What is the comparison of the fit of the curves in table 3 to those obtained by traditional RR reconstruction,  Can you demonstrate advantages?
The importance of resampling seams critical .  Can you say more advantages of smoothing over a finer matrix?
The mathematics are particularly difficult to follow because of the appalling  script used =  although this is easily remedied.
On the positive side it is pleasing to see a full physics working and an attempt to consider spill in from different background ratios
Finally, although I applaud every attempt made to produce a paper in a learned language, sadly this paper suffers from many grammatical errors and slips, and would benefit from a sympathetic revision from a native English speaker

Other points needing attention

Abstract  Background is not Cold if there is activity, but could be described as physiological

L197   missing Litres for volume

L245   home made = in house

L267  How did the VOI definition technique affect the measured volume of the spheres,  which is known?

Author Response

Please see the attachment, where the point-by point response are reported.

Thank you.

Round 2

Reviewer 2 Report

This paper has become more legible and the flow is much easier to follow.